# The effect of script reform on levels of orthographic knowledge: Evidence from alphasyllabary Malayalam scripts

Krithika Nambiar[1]☯*, Kiran Kishore[2], Pranesh Bhargava[1]☯*

1 Department of Humanities and Social Sciences, Birla Institute of Technology & Science Pilani, Hyderabad, India, 2 Centre for Neural and Cognitive Sciences, University of Hyderabad, Hyderabad, India

☯ These authors contributed equally to this work.

* p20190436@hyderabad.bits-pilani.ac.in (KN); pranesh@hyderabad.bits-pilani.ac.in (PB)

**Data Availability Statement:** The dataset (https://osf.io/nh6we/?view_only=041a1bd6e09949e2abdc6bf8b3be0288) is publicly available.

## Abstract

This study explores the idea of the two levels of orthographic knowledge, i.e. lexical and sublexical; in particular, how these levels are affected in the case of the Indian language Malayalam that went through a script reform in 1971. Through reading and writings tasks, we compare the performance of elderly participants who gained literacy in the traditional script (with complex ligatures), with younger participants who gained literacy in the reformed script (with simpler glyphs). Both the groups read text faster in reformed script indicating script simplification was beneficial. While writing, the elderly participants largely employed the traditional script and younger ones used the reformed script. The study provides proof from non-European alphabet that orthographic knowledge indeed has two independent but related levels. Although a change in script affects both the levels, sublexical one seems more resistant to change, possibly due to less opportunities to update it.

## 1. Introduction

Orthography, which literally means 'correct writing', deals with written forms of languages, including writing systems, script, and spelling and punctuation conventions. Primarily, it deals with the conventions mapping the spoken language with its script, which is the set of visual symbols employed to graphemically encode the spoken language [1]. As a field of study, it enjoys a unique position because it combines insights from such myriad fields of research as linguistics, typology, psychology, and reading-writing [2, 3]. From a social and policy perspective, orthography is important because it is the most indispensable component of literacy acquisition [4]. Despite its important influence on a number of fields of inquiries, orthography has faced some challenges. For example, there has been a lack of consistence in defining and measuring concepts in orthography [4, 5]. Apart from this, a significant number of studies are Anglo-centric or Euro-centric, conducted on limited scripts and spelling systems, leading to a noted lack of diversity in the data in the field [6, 7]. For example, see Vaid and Padakannaya [8] for an overview of different results from alphasyllabaries compared to alphabetic script. There is a lack of data from people who are biliterate and biscriptal (i.e. have the ability to read

**Funding:** The authors received no specific funding for this work.

**Competing interests:** The authors have declared that no competing interests exist.

and write two scripts) from non-Euro-centric languages with the shared mental lexicon of two different orthographies which are not alphabetic in nature [9].

This underscores the importance of conducting studies on orthography, but with other languages, scripts and writing systems, and correlating the results to see if the existing theories and frameworks continue to hold. This study, based on reading and writing of Malayalam script, tries to address the aforementioned issues, while referring to the concept of orthographic knowledge. We chose Malayalam not only because it is one of the little studied scripts from the Indian subcontinent, but also because spurred by the change in script few decades ago, it provides a unique opportunity to do a comparative analysis of two generations of Malayalam readers, each primarily literate in a different version of the Malayalam script. Given the old age of the users of the older script, this study capitalizes on a fast-closing window.

## 2. Malayalam orthography

### 2.1 Evolution of Malayalam script

Malayalam is a language with almost 38 million speakers, spoken primarily in the state of Kerala in India [10–12]. Malayalam orthography traces its roots to the ancient Brahmi script, making it an alphasyllabary system. However, because of its own rich history, and influence from neighbourhood linguistic communities, Malayalam script has undergone several changes, and accumulated writing conventions from different eras and communities. Over a span of few centuries, it was written with various variants of Brahmi, e.g., Vatteluttu (northern script), Grantha (scripture script), Koleluttu (rod script), Malayanma, and Aryaeluttu (elite script), etc. [13–15]. As a result, by the middle of 20th century, its orthography had become extremely diverse and complex. It not only had compound ligatures, but also different writing versions for the same graphemes influenced by the different prevailing styles. By 1950s, the prevailing Malayalam script had more than 1200 graphemes in the form of complex glyphs and ligatures, with sometimes more than one styles of forming graphemes for the same clusters of consonants and/or vowels [16].

In 1958, the state government of Kerala passed the Kerala Education Act, incorporating free and compulsory primary education within the state. A planned implementation of this allowed Kerala to become the first state in India to achieve near 100% literacy by the 1990s [16, 17]. As both the cause and effect of the improved literacy, Kerala experienced an ever-increasing demand for books, periodicals, pamphlets, and other printed material. Incorporating all the diverse ligatures of the Malayalam script in print and publications became a major challenge for the publication houses. At this time, the arrival of indigenously built, hence affordable, Indic script typewriters paved the way for distributed, faster, and longer lasting record keeping, thus boosting the print media circulation in other native languages [18]. However, the development of Malayalam typewriter faced the challenge of organizing its inventory of often redundant complex graphemes on the limited space of typical manual typewriters. A solution to this would have helped the large-scale offset printing as well.

### 2.2 Orthography reform in Malayalam

In 1971 the Kerala government acknowledged 'the unwieldy number of alphabets and signs in Malayalam' and the consequent labour involved in the process of printing [17]. A government committee found that by writing the consonants and diacritics separately rather than as complex ligatures the number of graphemes in Malayalam could be reduced by 75%, thus suggesting, (i) to discard the usage of complex conjuncts, (ii) and to detach the vowel notations from the consonants and conjuncts. Consequently, the state of Kerala passed the order, 'Malayalam

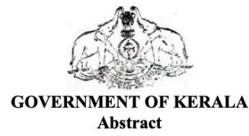

**GOVERNMENT OF KERALA**
Abstract

MALAYALAM SCRIPT-ADOPTION OF NEW SCRIPT FOR USE-ORDERS ISSUED

**EDUCATION 'P' DEPARTMENT**

G. O. (P) 37/71/Edn.          Dated, Trivandrum, 23rd March 1971.

Read: G.O. (P) 329/68/Edn.dated 11-7-1968

**ORDER**

The question of reducing the unwieldy number of alphabets and signs in Malayalam which consume much time and labour in the process of printing and typewriting, has been under consideration of Government for some time. In 1967 Government appointed a Committee with Shri Soornad P. N. Kunjan pillai, Editor, Malayalam Lexicon as convener to advise them on the question of reformation of Malayalam script. The committee in its report has made recommendations to reduce 75% of the total number of existing characters in printing and typewriting. The reformed Malayalam script recommended by the above Committee was revised with slight modifications by another committee appointed in 1969 to expedite the adoption of the new script for use. The recommendations of the above two committees in the matter of reformation of the Malayalam script are in brief as follows:

i.  ഉ, ഊ, ഋ, ര എന്നിവയുടെ മാത്രകൾ വ്യഞ്ജനങ്ങളിൽ നിന്നും വിടുവിക്കുക

ii. പ്രചാരം കുറഞ്ഞ കൂട്ടക്ഷരങ്ങൾ ചന്ദ്രക്കല ഉപയോഗിച്ച് പിരിച്ച് എഴുതുക.

2. In January 1971 a conference of Managing Editors of important newspapers in the State was convened to discuss the question of adoption of the new Malayalam script for use. The conference has recommended that the reformed script as revised by the committee might be adopted for use with effect from 15th April 1971 (Vishu Day).

3. Government have considered the question in detail and are pleased to accept the reformed Malayalam script as revised by the Committee for use. The new script will be adopted for all official purposes with effect from 15th April 1971 (Vishu Day). A booklet "ലിപിപരിഷ്കരണം" containing the details of the new script accepted by Government is appended to this G.O.

4. All newspapers and periodicals in Malayalam are requested to co-operate with Government in the implementation of the scheme and to adopt the new script from the stipulated date.

**Fig 1. Reform order snapshot: A short table describing the major proposals by the Kerala government in 1971 reform order, and some examples of the resulting differences in the traditional and reformed script.**

Script—Adoption of New Script for Use', in 1971 to discard the usage of ligatures to represent complex conjuncts, and to simplify them [14, 17, 19–21].

Through this order, a newer script called 'പുതിയ ലിപി' (read: puthiya lipi, meaning reformed script) of Malayalam came into effect on 15 April 1971. Henceforth, the older script shall be referred to as the traditional script (TS) and the newer version as the reformed script (RS). The 1971 order (Figs 1 and 2) brought down the number of graphemes from 1200 to a standardised 90, that includes 18 vowels and 39 consonants. This script reform initiated by the government was a major event in the evolution of the Malayalam orthography.

## 2.3 Orthographic knowledge and age

Orthographic knowledge is the interpretation of spoken language in print [4]. The specific graphemic form of any writing system include the letter shape, letter/grapheme configuration (Roman letters vs. Devanagari), syllable (or character) formation, diacritics or circumflexes, word constituents (simple or compound words), spatial relations, and syllabic format. These constituents yield considerable effects on mental processes of individuals such as their thinking, reasoning, stimuli recognition, concept formation and worldview [1, 3]. There is a

| No. | Proposals in 1971 Kerala Govt. Order | Traditional Orthography (TS) | Reformed Orthography (RS) |
|---|---|---|---|
| 1 | Detach the signs of vowels ഉ(u), ഊ(u:) and ഋ(ri) from the base grapheme. | i.കു (ku)  ii.കൂ (ku:)  iii.കൃ (kri) | കു (ku)  കൂ (ku:)  കൃ (kri) |
| 2 | Discard the usage of ർ (r) in the consonant sequence in the form of dot reph sign ( ˙ ). Alternate form ർ to be used. | i.ചക്ക (chark'ka)  ii.പ്രാത്ഥന (prārt'thana) | ചർക്ക (chark'ka)  പ്രാർത്ഥന (prārt'thana) |
| 3 | Discard the use of rare conjuncts by splitting them down into constituent consonant sequence separated by the virama sign. | i.ക്ത (kta)  ii.ശ്ച (shcha) | ക് ത (kta)  ശ് ച (shcha) |
| 4 | The consonant sign ർ (r) to be separated from the base grapheme using the alternate form �id . | i.പ്ര (pra)  ii.ക്ര (kra) | പ്ര (pra)  ക്ര (kra) |

**Fig 2. Summary of the government of Kerala's 'Malayalam Script—Adoption of New Script for Use' order– 23rd March 1971.**

relation, therefore, between other constituents of language, mental processes associated with language processing and its orthographic form or the script. The mental orthographic knowledge is a reflection of external linguistic system of a language.

Orthographic knowledge can be further divided into two types: lexical orthographic knowledge and sublexical orthographic knowledge [4, 5]. Lexical orthographic knowledge is the collection of the (parts of) words an individual already knows [22–24]. Sublexical orthographic knowledge is the collection of rules or patterns connecting a grapheme(s) to a sound or an affix, and rules of occurrence of graphemes in context of each other and word positions. The two concepts do not have a hierarchy despite what the names suggest, and are deployed in different reading/writing situations. Known and familiar words, which are in lexical orthographic knowledge, are fluently read and written with little cognitive efforts. On encountering novel words not in the lexical orthographic knowledge, sublexical orthographic knowledge is deployed [4]. The most used orthographic knowledge measure task in studies [25, 26] is the orthographic choice task. This task requires the participant to listen to the word spoken by the experimenter and then match it with the two written words in a word pair. The words in the word pair would have a correctly spelt word, an incorrectly spelt yet orthographically plausible version of the correctly spelt word (e.g., brain vs brane) or a homophone (e.g., pear vs pare). This is a judgement of the individual's lexical orthographic knowledge in the form of stored mental graphemic representations of words. The prevailing theories and research indicate that lexical and sublexical orthographic knowledge are independent but related concepts. [4, 27, 28].

Since orthographic knowledge is based on observation of patterns, it would be dependent on what script an individual was exposed to when s/he learnt to read and write, and the knowledge of its possible patterns formed over a period. Studies have indeed shown that the orthographic knowledge for a language form early on in a speaker's life, and is a function of exposure through education and reading habits [5, 29, 30]. Early reading habits are the

precursors of the later orthographic knowledge [31]. There is also evidence for orthographic knowledge modification in children when they learn new segments in their language [32–34]. These findings do indicate that children can acquire their mental representation of the script through their education, and later modify them based on reading or media exposure. However, there is not enough evidence connecting the progress of orthographic knowledge as age progresses beyond childhood.

This leads us to some interesting questions. Firstly, if the script of a language changes rapidly in a short time, then would there also be a difference in the orthographic knowledge of individuals who learnt to read and write before and after the change? This is not a straightforward question to answer, because an individual may have learnt to read and write in an old script to begin with but may be exposed to the new script in later years. It is not clear if and how the orthographic knowledge from the two scripts would interact with each other.

Secondly, would such a change reflect differently on the two levels of the orthographic knowledge? There are claims that say that sublexical orthographic knowledge gets formed first, and helps to develop the lexical orthographic knowledge as the learnt patterns are used in sounding out the novel words in children [29, 32, 34]. However, it is also claimed that the stored mental representations of words, i.e., lexical orthographic knowledge would improve the knowledge of patterns, i.e., sublexical orthographic knowledge. Thus, the two levels are different but interrelated, and if what affects one level would also affect the other remains to be determined.

Finally, does 'simplification' of script provide benefit of improved reading and writing efficiency? How does this benefit, if any, interact with exposure to the script? Since an individual may perform greater amount of reading activity than writing activity [35–37], it is possible that the two activities demonstrate different benefits.

Some of these questions can be tackled by running studies with people speaking the same language, but who were given literacy in different scripts; and/or people who were given literacy in one script but were exposed to a different script later in life. The historic Malayalam script reform move by the Kerala government in 1971 allows access to such people. Fifty-five years and older people in Kerala, the pre-reform group, had their elementary education in TS Malayalam before the reform of the government in 1971, whereas 54 years and younger, the post-reform group, had their elementary education after the reform in RS Malayalam. The years of a particular script exposure is a function of age as well. After the 1971 reform order by the government, there was a gradually increasing prevalence of RS in the print media and other publications such as administrative documents, court proceedings, textbooks, etc. A comparison of the orthographic knowledge, especially sublexical knowledge of the pre-reform and post-reform population would be interesting because the former had their entire elementary education in TS and later got exposed to print and other media in RS, while the latter had their education and media exposure both in a single script, i.e., RS.

## 3. Current study

The current study assessed the interaction of two different Malayalam scripts with lexical and sublexical orthographic knowledge.

We tested the interaction of scripts with lexical orthographic knowledge using a reading task. The interaction was measured as the processing demand (i.e. how much of mental resources are engaged in doing a task) involved in reading the scripts. The rationale is that if there is sufficient lexical orthographic knowledge of the words, e.g., mental graphemic representation of the presented words, then there is little cognitive effort in recognizing the word

while reading, as a result, the reading time should be faster as compared to the cases where there is no lexical orthographic knowledge [4, 28].

Since the script in which the participants had their elementary education would comprise the lexical orthographic knowledge, we hypothesised that reading in that script would lead to lower processing demand and faster reading times. This means that for the pre-reform group, reading sentences in TS as compared to RS would induce less processing demand, which would be reflected as shorter reading time. Similarly, the post-reform group would experience less processing demand, and hence shorter reading times while reading sentences in RS.

We also examined the effect of script reform on sublexical orthographic knowledge by observing the writing skills of participants. Sublexical orthographic knowledge deals with the rules connecting spoken sound to written grapheme and how graphemes occur next to each other, hence would be activated while writing dictated speech. We hypothesised that, just like the lexical orthographic knowledge, the sublexical orthographic knowledge of the participants would enshrine the script that they learnt in their elementary education [4, 5, 23]. Thus, it is this script they would use when asked to produce written material. This means, the pre-reform participants would choose to write in TS, and the post-reform participants would choose to write in RS.

However, since the reform was implemented almost 40 years ago, and print media had to adapt to RS, all the participants would have been exposed to RS in print. Even though reading the printed words deals primarily with lexical orthographic knowledge, it is possible that this significant exposure to the RS script had an impact on sublexical orthographic knowledge of the pre-reform group. This could be tested through observing the script that individuals subconsciously choose to write.

Any discrepancies in the usage of the script while reading or writing Malayalam by the participants from a particular age group would tell us about the impact of the script reform on orthographic knowledge for individuals. This would elucidate how script itself and the changes induced to it would affect the processing demands of individuals across age groups, and whether their elementary training and exposure to one script would have an impact on their reading and writing skills, shedding more light on orthographic knowledge, using data from a non-European language.

## 4. Method

### 4.1 Participants

The study was carried out in the state of Kerala in India. A total of 60 first language Malayalam speakers were recruited through the word of mouth and referrals. The participants were divided into two groups (30 each) based on age, and hence when they received their elementary education. The script reform order by the government was implemented in 1971. The pre-reform group, 55 years and above in age, were participants who received their elementary education in TS Malayalam before 1971 and the post-reform group, 54 years and below in age, were participants who received their elementary education in RS Malayalam post 1971 reform order.

Participants (Females = 39, Males = 21) ranged in age from 17 to 80 years with average 20.3 years of education. The participants had normal-to-corrected vision. As part of our pre-test measures the pre-reform group of 55 years and above participants were screened for cognitive status using the Addenbrooke's Cognitive Examination—Malayalam (M-ACE) test, which is the Malayalam adaptation of the Addenbrooke's Cognitive Examination [38–40]. All participants from this selected group scored more than 88 points out of 100 on M-ACE with no indication of cognitive impairment.

## 4.2 Technique

Self-paced reading (SPR) is a computer-based psycholinguistic technique replicating real time language comprehension process through tasks similar to normal reading (Mitchell & Green, 1978). The words are presented one at a time on a computer screen, at a speed controlled by the reader, hence the name.

We implemented a non-cumulative SPR technique, in which all the words of the sentence are available on the screen but masked, i.e., not visible. Pressing a keyboard key unmasks only one word. Each subsequent keypress unmasks the next word while masking the previous one again. Thus, on successive keypresses, the words appear and disappear one-by-one, from left to right, in linear succession. The time taken in milliseconds to read the unmasked word, measured as the time between two successive keypresses, was called the reading time (RT) per stimulus word. The RT is indicative of the demands on the processing capacity, i.e. longer RT indicates higher demands on sentence processing or lower resources available. Being inexpensive, easy to implement, and portable, SPR technique was suitable for this study because the research had to be carried out at the homes of elderly participants.

Sentences were presented through a user interface, custom-made for the study on Linger software (v2.88) which is a standard Tcl/Tk application for performing SPR [41], and because it supports both the Malayalam scripts (TS and RS). The font size was 45, and the font type was AnjaliNewLipi and AnjaliOldLipi for RS and TS, respectively.

In the second part of the study, each participant was orally dictated 5 sentences by the experimenter. These sentences were different from the previously mentioned 50 sentences. The same 5 sentences were dictated for all the participants. Each sentence comprised words which had 6–7 instances of glyphs from the 4 major reforms mentioned in Fig 2. This resulted in 30–35 instances each of testable glyphs from the written data of the participants.

## 4.3 Sentence material

For the present study, complex glyphs are considered to be the glyphs from TS which are formed when (i) two or more consonants are conjoined, or (ii) when vowel notations are attached to such consonant conjuncts or single consonants (Fig 2). A set of 50 Malayalam sentences in TS (total 323 words) was created, ensuring that each sentence had at least four instances of these complex glyphs. Within a sentence, the complex glyphs could be distributed across unique words, or two or more complex glyphs could occur within a single word. The latter case is classified as the Complex Word Type. Each sentence from the TS experiment set, thus had at least 4 complex glyphs, but may or may not have the Complex Word Type. The TS set had total 59 words in the Complex Word Type category. Each sentence was meaningful as a whole, but the sentences were created in a way to avoid the words to be guessed from the context. A corresponding set of 50 sentences in RS was obtained by replacing the complex glyphs of TS script with glyphs in RS script (Refer Fig 3 for an example). Each participant was presented with both the sets one after the other, i.e., total 100 sentences.

*TS:* ഗുഗിൾ നിയന്ത്രണം എത്ര പ്രവാസികളെ വലയ്ന്തു.

*RS:* ഗൂഗിൾ നിയന്ത്രണം എത്ര പ്രവാസികളെ വലയ്ക്കുന്നു.
   */Google  niyanthranam ethra pravaasikale     valaykkunnu./*

meaning: *Restriction on Google affected the migrants*

**Fig 3. An example of a sentence in TS and RS.**

## 4.4 Procedure

Participants signed an informed consent form, followed by filling the demographic form, and undergoing the M-ACE test. Following this, the SPR task was administered by the experimenter on a 14-inch Lenovo laptop (AMD Ryzen 7). Participants sat approximately 80 cm from the laptop screen in a quiet place in their house. An absence of obvious distractions and human interruption was ensured during the experiment. The presentation of stimuli was controlled through the user interface. RT as duration between two keypresses was measured in milliseconds (refer to Fig 4 for the schematic representation of the experimental set-up).

Each participant read the same 50 Malayalam sentences once in RS and once in TS. The order of the script, as well as the order of the sentences were randomised across participants. An instruction page appeared for both the sets before the participants began the reading task. Additionally, the experimenter orally described the procedure to the participants. A short training session was provided to familiarize the participant with the interface and the requirements of the task, without disclosing the use of or significance of the different scripts. To ensure the participants really read the words and were not skimming through the sentences, they were instructed to read the words aloud at speaking voice level. The reading activity was audio-recorded and transcribed for later confirmation. An automated break was programmed to be enforced after 25 sentences. The entire reading task lasted for approximately 30 minutes per participant. The study was conducted in accordance to the guidelines by the Institutional

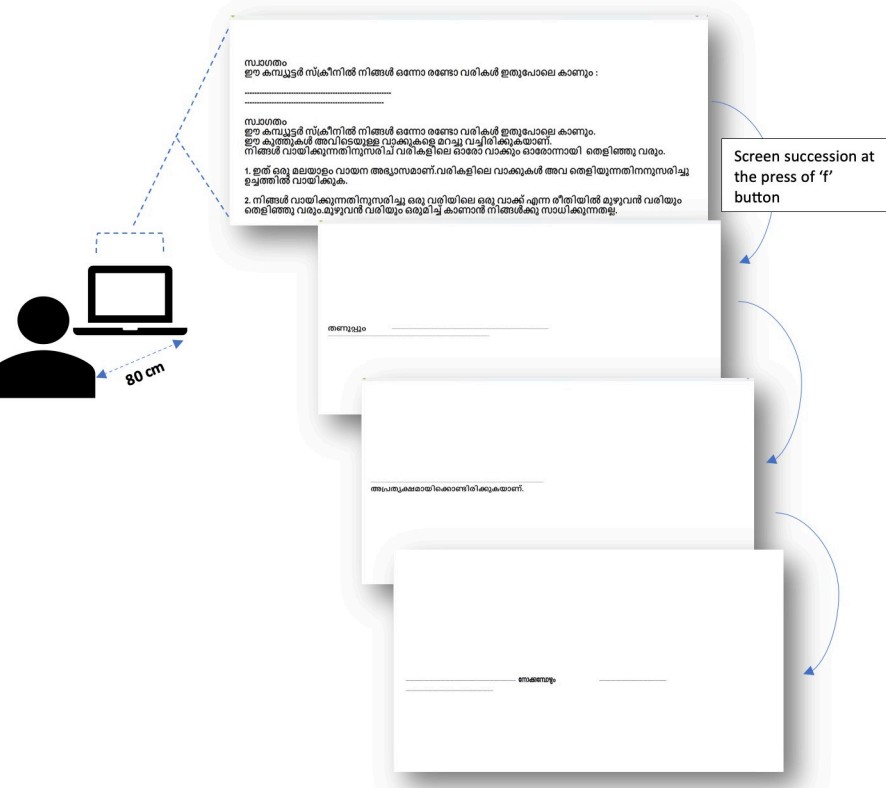

**Fig 4. Schematic representation of the self-paced reading experiment.** The participant is seated 80cm away from the laptop screen. The participant sees the instruction page, followed by the sentences in TS (or RS), each sentence appearing one word at a time.

Ethics Committee of Birla Institute of Technology and Science, Pilani—Hyderabad Campus, India (Protocol Code—BITS-HYD/IHEC/2022/04).

# 5. Results

For the reading test, the RT in milliseconds for each sentence was summed up as cumulative RT for that sentence. The cumulative RT of the sentences was converted into seconds and averaged across participants for each of the group. The values of the number of written glyphs were also averaged across participants for each of the two age groups. This data is presented in Table 1.

There were total 6000 observations (60 participants * 50 sentences * 2 scripts i.e., TS and RS) for the reading time analyses. Gelman and Hill [42] recommend using the log-transformed values of RT because the untransformed reading times fail to meet the assumptions of additivity and linearity. To transform the distribution to normality, the natural logarithm of RT (logRT) for each word read by the participants was calculated. Further analysis of reading time was done on these log-transformed values. Refer to the QQ plot in Fig 5.

The time taken to read each word with or without the complex glyphs was analysed using the open-source version of RStudio, integrated development environment (IDE) for R (64-bit, version 3.5.1). All the packages that were used within R were installed through the cloud library of R-Cran. For plotting the graphs from within R, Rcmdr package version 2.5–1 was used. A two-by-two design linear mixed model by REML ('lmerMod' package in R) was used.. The fixed effects included the script type (TS and RS) and the age group (pre-reform, i.e., 55 years and above, and post-reform, i.e., 54 years and below). The random effect structure consisted of the random by-participants and by-item (word in a sentence) intercepts. The dependent variable was Log (RT). Degrees of freedom, and consequently, the p-values were estimated using Type II Wald chi-square tests.

Table 2 provides the output of the model. The model shows a main effect for script type and age group. The participants in the post-reform, i.e., 54 and below age group took more time to read the sentences in TS (mean = 50.73 second, SD = 19.7) than to read the sentences in RS (mean = 37 second, SD = 14.35). Similarly, the participants in the pre-reform, i.e., 55 and above age group also took more time to read sentences in TS (mean = 56.03 second, SD = 17.96) than to read sentences in RS (mean = 43.36 second, SD = 16.37). There was also a script type * age group interaction. This suggests that irrespective of the age group, the time taken to read sentences was longer in TS than RS. This indicates that reading words in TS induced greater processing demand for both the age groups.

Refer to Fig 6 for the plot of log-transformed means of RT. Note that the y-axis does not begin at zero in this figure.

In order to further evaluate the significant differences in the two scripts and their processing, the reading times of words with complex glyphs were separately analysed., i.e., TS and RS. For each participant, the RT in milliseconds for reading the complex glyph words was aggregated, and averaged over the number of participants of the same age (Refer to Fig 7). It was

**Table 1. Mean (and standard deviation) among script types and age groups for reading time in reading task and the distribution of written glyphs in writing task.**
The mean of the cumulative RT per sentence is expressed in seconds. The number of written glyphs for writing task totals to 35 for each script.

| | Mean RT with Std. Dev. (in seconds) | | *Mean Written Glyphs (SD)* | |
|---|---|---|---|---|
| | Post-reform (54 yrs. and below) | Pre-reform (55 yrs. and above) | Post-reform (54 yrs. and below) | Pre-reform (55 yrs. and above) |
| *Reformed Script* | 37 (14.35) | 43.36 (16.37) | 22.17 (12.66) | 4.39 (8.15) |
| *Traditional Script* | 50.73 (19.7) | 56.03 (17.96) | 12.83 (12.66) | 30.61 (8.15) |

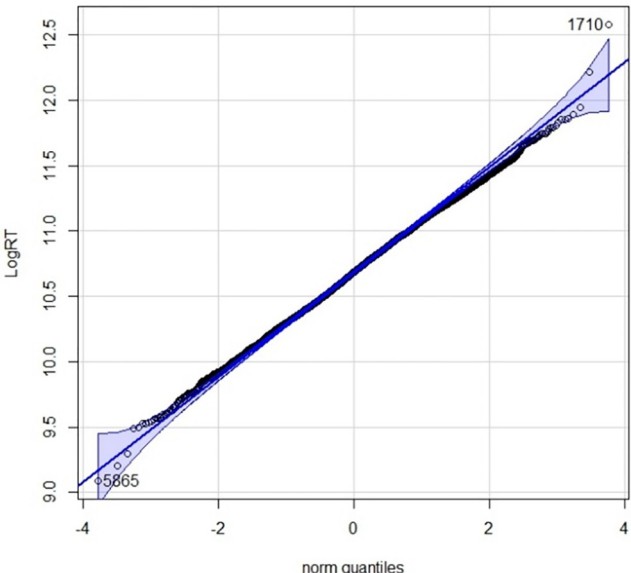

**Fig 5. Quantile-Quantile plot of RT data points of the total 6000 observations showing normality in distribution.**

then converted into seconds. The average RT of words with complex glyphs in the case of TS were found to be significantly higher (mean = 8.4 second, SD = 1.96) than that of RS (mean = 6.3 second, SD = 1.95). Refer to Fig 7 for a detailed script-wise RT distribution for these words with complex glyphs across age. There is a significant effect of the words with complex glyphs written in TS on RT (Table 2, word type, p <0.001).

The sentences written by the participants from the two age groups were analysed. The mean of the distribution of each script type in writing the glyphs was calculated (i.e., what proportion of the 35 glyphs was written in RS vs. TS by each group). Refer to the data provided in Table 1. Participants in the post-reform group wrote more frequently in RS (mean = 22.17

**Table 2. Outputs of linear mixed effects model for the reading time data.**

| Fixed effects | Est | Std. Er. | t value | Variance | Std. Dev. | χ2 | Pr(>\|χ2\|) |
|---|---|---|---|---|---|---|---|
| (Intercept) | 10.446 | 0.055 | 190.747 | - | - | - | - |
| Script.Type | 0.323 | 0.005 | 54.519 | - | - | 5171.317 | < 2.2e-16 *** |
| Word.Type | -0.123 | 0.028 | -4.376 | | | 19.1482 | 0.0000121 *** |
| Age.Group | 0.167 | 0.068 | 2.439 | - | - | 4.257 | 0.03908 * |
| Script.Type X Age.Group | -0.052 | 0.008 | -6.294 | - | - | 39.619 | 3.086e-10 *** |
| *Random effects* | | | | | | | |
| Participant (Int.) | - | - | - | 0.069 | 0.264 | - | - |
| Word (Int.) | - | - | - | 0.029 | 0.17 | - | - |
| Residual | - | - | - | 0.025 | 0.159 | - | - |

Significance codes:

*** < 0.001,

** < 0.01,

* < 0.05,.< 0.1

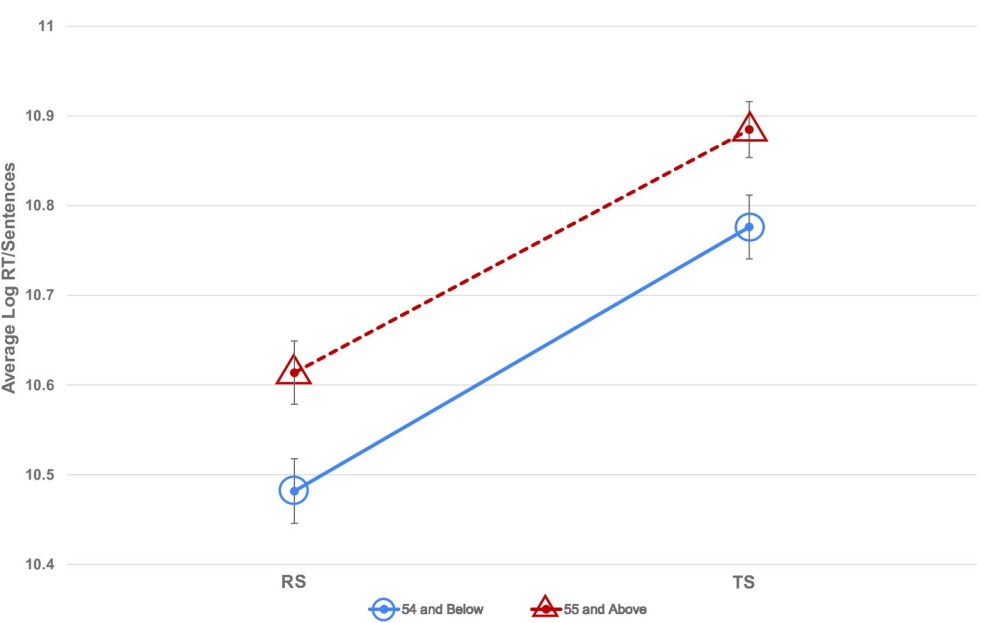

**Fig 6. Plot of log-transformed means of reading time (RT), and the two-way interaction between the script types (RS and TS) and age group (54 and below; 55 and above).** The y-axis does not begin at zero.

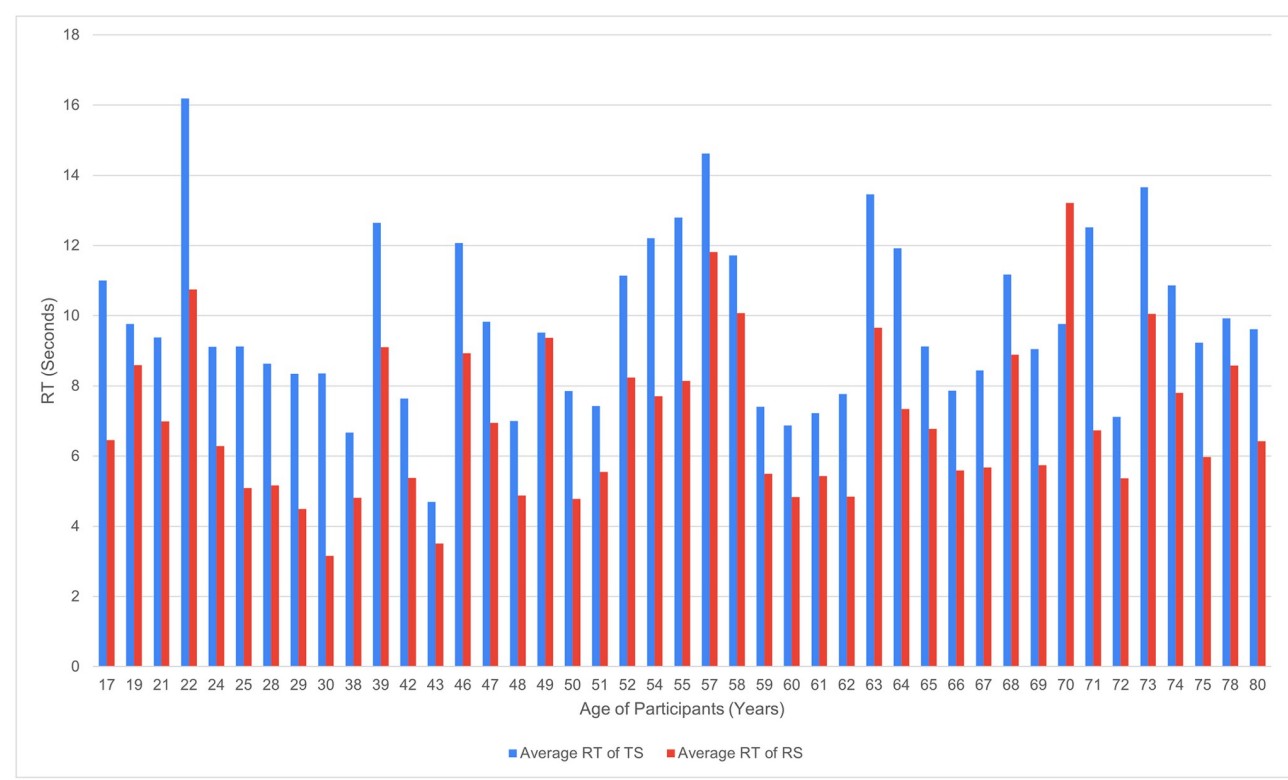

**Fig 7. Script-wise distribution of RT taken to read words with complex glyphs by the participants across age.**

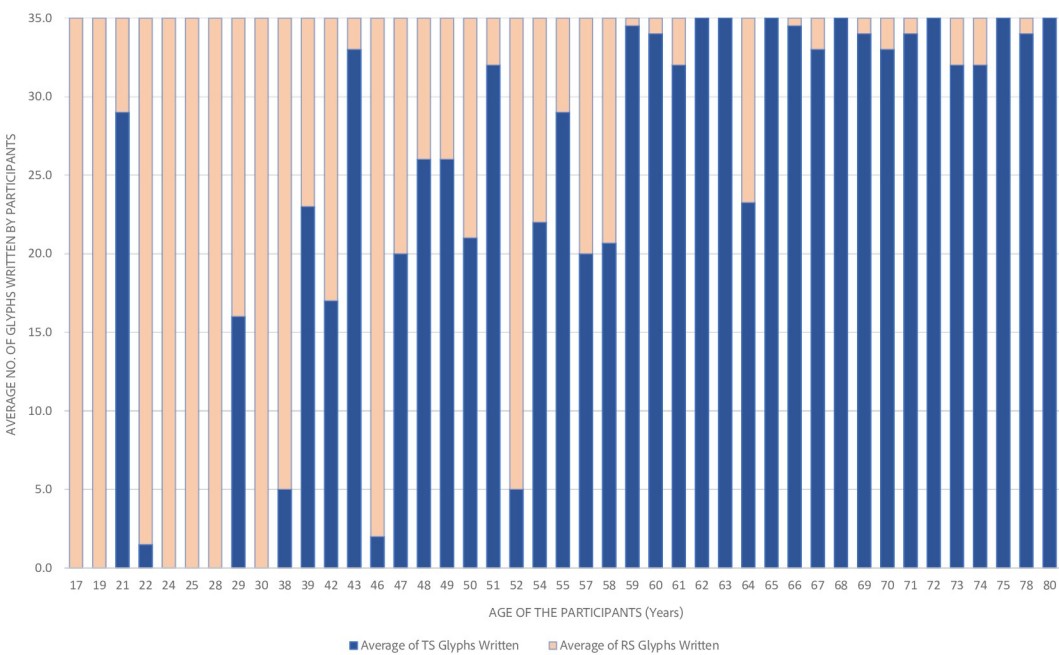

**Fig 8. Script-wise distribution of the number of glyphs written by the participants across age.**

glyphs, SD = 12.66) than TS (mean = 12.83 glyphs, SD = 12.66). The participants in the pre-reform group on the contrary used more glyphs from the TS (mean = 30.61 glyphs, SD = 8.15) than glyphs from RS (mean = 4.39 glyphs, SD = 8.15). Refer to Fig 8 for the share of the two scripts in the total number of glyphs written by the participants as a function of age. We checked the correlation between the age group of the participants and the RS glyphs written by them using the Pearson's product-moment correlation. The $r$ value was -0.73. The $r$ value between the age group of participants and the TS glyphs written by them was 0.73. This indicates that the age group of participants has an inverse correlation with RS glyphs and a positive correlation with TS glyphs.

## 6. Discussion

This study was conducted to see how a change in script interacts with orthographic knowledge of the users of the script. This was achieved by comparing the reading and writing measures of participants who had their elementary education in TS Malayalam (pre-reform, i.e. 55 years and above) with those who had their preliminary education in RS Malayalam (post-reform, i.e. 54 years and below). Whereas the pre-reform group had ample exposure to TS, the post-reform group had exposure to mostly only RS.

First off, we found that both the groups were able to read both the scripts, though with some difference in performance as shall be discussed later. It may be noted that though the reform was passed in the 1971, the books and print media in TS that were in circulation before the reform were not banned and have continued to be in circulation even after the reform. For example, in spendthrift poor and middle-class Indian families, the school textbooks once bought may be used for home-based instruction year-after-year by younger learners within the family. The literary books and religious literature may be preserved for reading across years as well. Thus, print material in TS may have been in circulation alongside the material published

in the RS. This might explain why even the younger people were able to read in TS, albeit slowly.

We had hypothesised that the lexical orthographic knowledge should primarily be composed of the script in which the participants had their elementary education, and this lexical orthographic knowledge would have an impact on script processing, i.e., reading. Based on this hypothesis, we expected that the pre-reform group participants would experience less processing demand when reading sentences in TS compared to RS, which should be reflected as shorter reading time (RT) for sentences in TS compared to RS. Similarly, the post-reform participants would experience more processing demand, reflected as longer RT while reading sentences in TS as compared to RS. However, against the expectation of the hypothesis, our results of the self-paced reading test show that the participants of both the groups were faster in reading sentences in RS as compared to the sentences in TS. Although the elderly pre-reform group took longer time than the younger post-reform group, which could be explained on the basis of general slowdown of the executive functions in the elderly participants [43], the difference between RS and TS persisted within both the groups.

The fact that both the groups read faster in RS indicates that there was less processing demand in reading words in RS script. One indication of this is that the lexical orthographic knowledge for both the groups is largely composed of mental graphemic images of the words in RS, making the retrieval faster. This makes sense for the post-reform group who had received their elementary education in RS and not in TS. But for the pre-reform group, who had their education in TS, the reading results imply that either RS has completely replaced TS from the lexical orthographic knowledge, or more likely, RS co-exists with TS in lexical orthographic knowledge, and somehow RS has a significant presence, closer to retrieval. This could be true because reading is a skill that is practice-based, and hence would be sensitive to exposure. Since the print media and educational textbooks changed the fonts to RS from TS after the reform order, the reading exposure must be in RS.

An additional or alternate explanation is that RS is inherently easier to read as compared with TS, resulting in lower processing demand and shorter reading times. The consonant-vowel and consonant-consonant clusters in TS form a unique glyph different from either of the constituent letters. The letters are written as separate entities in RS. Because of this, RS is likely to have lower complexity making it easier to read as compared to TS. An indication of evidence comes from Liu et al. [43] who found that elderly have lower thresholds for crowding of the letters in reading tasks. This seems like a tenable explanation, which can be further strengthened through the tests of discriminability.

The writing task was provided to observe the interaction of the scripts with sublexical orthographic knowledge. The task of writing the dictated speech would have activated the rules connecting sounds to graphemes, and the occurrence of graphemes in each other's context, which is what sublexical orthographic knowledge deals with. The participants were asked to write the dictated sentences but were not overtly asked to pick any script. It was expected that when asked to produce written material, participants would use the script they had learnt in their elementary education, *viz.* the pre-reform participants would write in TS, and the post-reform participants would write in RS. Following the expectation, we found a clear relationship between the group type and the kind of the glyphs written. The pre-reform group wrote primarily using the TS glyphs, whereas the post-reform group wrote primarily using the RS glyphs (Fig 8). This confirms that sublexical knowledge is indeed formed with the script one learns in the elementary education. This is further confirmed by the correlation results. For the pre-reform group, there is an inverse correlation with RS glyphs and positive correlation with TS glyphs. This implies that the group type uses sublexical knowledge the script in which they had elementary education to write in a dictation task.

The writing task had two outliers from the post-reform group. A 21-year-old participant and a 43-year-old participant wrote with majority TS glyphs instead of the expected RS glyphs (Fig 8). Both the participants reported taking keen interest in the Malayalam language and its rich literary history, prompting a self-study of the TS glyphs using books printed in TS before the 1971 reform. On the other hand, a 64-year-old person in the pre-reform group wrote a large number of RS glyphs. This participant is a school teacher who teaches young pupils, and employ RS for the daily teaching activities. This shows that a conscious effort and practice may help a person override the script acquired through their elementary education and thus reorganize their sublexical orthographic knowledge at a personal level.

One would expect both the lexical and sublexical orthographic knowledge to be composed of the same script. This does seem to be the case with the post-reform group. However, with the pre-reform group, the reading results indicate the lexical orthographic knowledge to be composed of largely RS, whereas the writing results indicate the sublexical orthographic knowledge to be composed of largely TS. Although the lexical orthographic knowledge seems to be updated to RS, why hasn't the writing and sublexical knowledge updated to RS in this group? Is writing skill not exposure dependent?

The two persons in the post-reform group, as discussed earlier, have 'trained' themselves to use TS instead of RS in writing, through exposure to older books. Additionally, one person in the pre-reform group shows a greater use of RS than others in her group, due to using RS as required by her profession of school teaching. This indicates that writing skill may actually be exposure dependent after all, and it should be possible to update it. The question is, why have the rest of the pre-reform group not updated the written script to RS? One of the explanations is that writing is a production-based skill. In order to update the writing skill, one needs to practise writing it. However, with the advent of the typewriters, computer-based text editing, larger circulation of printed material, audio-visual means of communication (e.g., telegraph, emails, etc), there is lesser opportunity for language users to write using RS, and hence update their sublexical orthographic knowledge.

This difference in the reading and writing results in the pre-reform group is a strong indicator that the orthographic knowledge indeed has two independent and separate levels. The sublexical orthographic knowledge from the results seems to be more rigid than lexical orthographic knowledge as it does not update with years of exposure as seen with the pre-reform group participants. This also implies that reading and writing functions differently in case of scripts, and like the orthographic knowledge should be tested at two levels each in case of lexical and sublexical knowledge.

The dichotomy similar to the two levels of the orthographic knowledge has been posited in the Dual-Route Theory of reading aloud [44, 45]. The theory suggests that while reading aloud, known words can be recognized by scanning the whole words and matching them with the mental database [46]. This is akin to the mental graphemic representations of the lexical orthographic knowledge. For unknown words, the reader takes the nonlexical route, which involves sounding out the word using the letter-sound rule system [47, 48]. In the SPR test deployed in the study, to ensure the participants were not skimming through the sentences, they were instructed to read the words aloud at speaking voice level. For those words, where the participants took longer time to read, which were unknown to them, they could have taken the nonlexical route of the Dual-Route Theory. Similarly, the shorter reading time taken by the participants while reading aloud could be the result of them matching the known words as an entire graphemic representation with their mental database. Thus, our findings fit well with the dual-route theory, except that the dual-route theory deals only with reading aloud, without referring to writing skills, whereas the present study explores that aspect as well.

## 7. Conclusion

Malayalam went through a script reform in the early 1970s, leading to simplifying ligatures and complex glyphs into simpler ones. The number of unique Malayalam graphemes went from almost 1200 in traditional script (TS) to almost 90 graphemes in reformed script (RS) following the reform, leading to more efficient production of print material and media. People brought up before the reform had their elementary education in TS, whereas the ones brought up later had it in RS. This allows for a unique, but continuously diminishing opportunity (because of the aging population), to study the effect of script change on orthographic knowledge for a language. The study indicates that:

(i) Orthographic knowledge indeed seems to have two independent but related levels.

(ii) Reading skills are exposure dependent, hence the lexical orthographic knowledge may get updated provided the condition of exposure.

(iii) Writing skill also seems exposure dependent, but there are fewer opportunities for language users to update their writing skills in modern times. Because of this, the sublexical orthographic knowledge seems more rigid than the lexical orthographic knowledge.

(iv) RS seems to induce less processing demand. This is a relevant result because the Kerala government had not considered the cognitive aspects of the change, such as its possible effects on orthographic knowledge of its speakers.

Gradually, it will be more and more difficult to do this kind of study again as the unique population who have been exposed to the two scripts continues to get diminished everyday due to cognitive impairment, old age, and death. The current study finds further relevance in the fact that in a bid to restore the rich cultural heritage of the classical Malayalam language, Kerala government is mulling over at least partially bringing back certain aspects of the old script [49–51].

The existing consensus of a lexical and sub lexical orthographic perspectives are mostly Anglo-centric in nature with analyses mostly with roman script or European alphabets. The current study, therefore, is a step towards bringing universality in the claim of existence of the levels of orthographic knowledge, using a lesser studied script from the Indian subcontinent.

## Author Contributions

**Conceptualization:** Krithika Nambiar, Kiran Kishore, Pranesh Bhargava.

**Data curation:** Krithika Nambiar, Kiran Kishore, Pranesh Bhargava.

**Formal analysis:** Krithika Nambiar, Kiran Kishore, Pranesh Bhargava.

**Investigation:** Krithika Nambiar, Pranesh Bhargava.

**Methodology:** Krithika Nambiar, Pranesh Bhargava.

**Project administration:** Pranesh Bhargava.

**Resources:** Krithika Nambiar, Kiran Kishore, Pranesh Bhargava.

**Software:** Krithika Nambiar, Kiran Kishore.

**Supervision:** Pranesh Bhargava.

**Validation:** Krithika Nambiar, Kiran Kishore, Pranesh Bhargava.

**Visualization:** Krithika Nambiar, Kiran Kishore, Pranesh Bhargava.

**Writing – original draft:** Krithika Nambiar, Pranesh Bhargava.

**Writing – review & editing:** Krithika Nambiar, Kiran Kishore, Pranesh Bhargava.

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
