## [Decision Letter · Decision Letter 0]

27 Mar 2023

PONE-D-22-29580The Effect of script reform on levels of orthographic knowledge: Evidence from alphasyllabary Malayalam scriptsPLOS ONE

Dear Dr. Bhargava,

Thank you for submitting your manuscript to PLOS ONE. After careful consideration, we feel that it has merit but does not fully meet PLOS ONE’s publication criteria as it currently stands. Therefore, we invite you to submit a revised version of the manuscript that addresses the points raised during the review process. Please submit your revised manuscript by May 11 2023 11:59PM. If you will need more time than this to complete your revisions, please reply to this message or contact the journal office at plosone@plos.org. Please include the following items when submitting your revised manuscript:A rebuttal letter that responds to each point raised by the academic editor and reviewer(s). You should upload this letter as a separate file labeled 'Response to Reviewers'.A marked-up copy of your manuscript that highlights changes made to the original version. You should upload this as a separate file labeled 'Revised Manuscript with Track Changes'.An unmarked version of your revised paper without tracked changes. You should upload this as a separate file labeled 'Manuscript'.If applicable, we recommend that you deposit your laboratory protocols in protocols.io to enhance the reproducibility of your results. Protocols.io assigns your protocol its own identifier (DOI) so that it can be cited independently in the future. For instructions see: https://journals.plos.org/plosone/s/submission-guidelines#loc-laboratory-protocols. Additionally, PLOS ONE offers an option for publishing peer-reviewed Lab Protocol articles, which describe protocols hosted on protocols.io. Read more information on sharing protocols at https://plos.org/protocols?utm_medium=editorial-email&utm_source=authorletters&utm_campaign=protocols.

We look forward to receiving your revised manuscript.

Kind regards,

Iftikhar Ahmed Khan

Academic Editor

PLOS ONE

Journal Requirements:

2. Please change "female” or "male" to "woman” or "man" as appropriate, when used as a noun (see for instance https://apastyle.apa.org/style-grammar-guidelines/bias-free-language/gender).

Additional Editor Comments:

Thank you for submitting your manuscript #PONE-D-22-29580 entitled "The Effect of script reform on levels of orthographic knowledge: Evidence from alphasyllabary Malayalam scripts" to PLOS One.

Based on reviewer reports, I recommend minor revisions in the manuscript. Please find the reviewer comments below in the email. I urge you to complete revision as soon as possible so that we can take our final decision.

Regards,

Reviewer 1 comments:

The study carried out is technically sound with adequate supported research. The argument is well established through literature review and structured methodology. References substantiate the central idea of the paper.

The following are some of the concerns that needs to be reviewed:

In the data provided (uploaded Excel sheet), the classification of word type – Complex/ Non-Complex is confusing. In the study, you have mentioned that;

“The sentences had total 59 words with complex glyphs, such that each sentence had at least four instances of the complex glyphs.”

Please elaborate the premise you used to characterize certain glyphs as complex glyphs.

Additionally, the classification in terms of Word Type requires further justification as in some sentences, 1 or 2 words are classified as ‘Complex’ Word Type while in other sentences, there are NO words that are classified as ‘Complex’ Word Type.

Additional concerns with LINE number:

Line 350-353

“We checked the correlation between the age group of the participants and the RS glyphs written by them using the Pearson's product-moment correlation. The r value was 0.73. The r value between the age group of participants and the TS glyphs written by them was 0.73.”

Please add the interpretation of this particular analysis.

Line 453-460

The reference is on point. So, it can be backed up with further explanation. If possible, please elaborate on how the present study explores the mentioned aspect (Line 460)

Line 131-132 (In-Text citation)

“Studies have indeed shown that the orthographic knowledge for a language form early on in a speaker’s life, and is a function of exposure through education and reading habits (Apel, 2011; Apel et al., 2006; Masterson & Apel, 2010).”

All other references citation in-text has been numbered with the exception of this.

Line 611-612

Lond Rev Educ

The abbreviation of this journal in the journal homepage is mentioned as “The London Review of Education (LRE)”

Journal homepage: https://uclpress.scienceopen.com/collection/2122ff6c-4fcf-4054-a1f0-3be2b1ac2eae#:~:text=The%20London%20Review%20of%20Education,sectors%20and%20phases%20of%20education.

Journal article: https://uclpress.scienceopen.com/hosted-document?doi=10.1080/13603110600574454

Please revise accordingly.

Figure 2

In Figure 2, the schematic representation of the self-paced reading experiment shows the instruction page followed by a sentence in RS. If possible, please add a sentence in TS also.

Reviewer 2:

The paper presents a very interesting processing study that has a cultural as well as historical aspect to it. I recommend it for publication. The study looks at whether orthographic knowledge is affected by changes (institutionalised reform) in the standard orthography of a language. Specifically, the paper looks at how the Malayalam script reform in 1971 affected the lexical and sublexical orthographic knowledge in the users of the script. The chronology of the reform allowed the authors to have a pre-reform and a post-reform group. The expectation was that the pre-reform group would take longer to read the reformed script (RS) and the post reform, the traditional script (TS). The participants were also given a production task, specifically a writing task, where they were asked to write down sentences that were dictated to them. I must mention at this point that the script reform was intended as a solution to 'redundant complex grapheme' and an absence of space in the typical manual typewriter of the time. It was predicted that the pre-reform group would choose to write in TS and the post-reform group in RS. Findings showed that both groups were faster in reading RS; and for the writing task, the pre-reform group continued to use TS but the post-reform group wrote in RS.

The results are very interesting, and the analysis provided was enlightening. The reason for both groups being faster was attributed to longer exposure to RS script post 1971. This is taken to imply that lexical orthographic knowledge is flexible depending on exposure. Then the question is, why is this pattern not observed in the writing task? This is where sublexical orthographic knowledge comes into play; the study suggests that sublexical rules ‘seem’ rigid, although their rigidity may be due to a lack of writing practice in the age of digital media. Even if you are a fluent reader of RS, if you learnt TS in school, you might choose TS. This is because writing is also a production task, which may be affected by writing practice. The correlation between group type and script type is taken as further evidence for this type of analysis. An important conclusion that the paper arrives at is the confirmation that orthographic knowledge consists of two independent but related levels: lexical and sublexical orthographic knowledge.

Additional interesting points from the paper:

1. The paper provides a good overview of certain characteristics of Malayalam orthography that can have consequences in terms of language processing, such as complex glyphs. Additionally, it also provides the historical context and contemporary relevance for orthography reforms.

2. The paper also explains how experimental methodologies such as self-paced reading tasks and writing tasks differ; the former is a perception-based task whereas the latter is a production task. The suggestion that a lack of exposure in the form of practice affects production tasks, as they generally incur more processing load is significant.

Points to improve/Suggestions for further research:

In section 6 (lines 414-415), the paper states that the correlation between group type and script type confirms that sublexical knowledge is determined by the script one learns in elementary education. However, does this correlation imply causation or confirmation? This question is significant because the later discussion mentions that sublexical orthographic knowledge could be restructured through continued exposure in the form of writing practice. There seems to be some conflict in clarity there.

In addition to writing being a production skill, it also differs from reading in terms of modality. Writing also has a motor processing component to it in addition to retrieval of orthographic representations. The paper briefly mentions age being a possible factor in writing speed in older participants. Is it possible for this difference in modality to have other consequences when it comes to the choice or preference of script type apart from exposure/writing practice?

It would have been helpful if lexical and sublexical orthographic knowledge were explained using examples/language data.

Similarly, more explanation for certain memory and processing related terms such as retrieval would be helpful for a reader who may be new to the area.

Is it possible that changes made in the script reform were also keeping in view that these revisions would simplify the script in terms of the cognitive load on the user? The paper passingly suggests otherwise. However, do you think it would be right to assume that reforms with such pragmatic intentions would naturally result in a simpler yet more efficient orthographic system?

I would also like to alert the authors of some minor proof-reading errors that need to be corrected.

Reviewers' comments:

Reviewer's Responses to Questions

**Comments to the Author**

1. Is the manuscript technically sound, and do the data support the conclusions?

Reviewer #1: Yes

Reviewer #2: Yes

2. Has the statistical analysis been performed appropriately and rigorously? 

Reviewer #1: Yes

Reviewer #2: Yes

3. Have the authors made all data underlying the findings in their manuscript fully available?

Reviewer #1: Yes

Reviewer #2: Yes

4. Is the manuscript presented in an intelligible fashion and written in standard English?

Reviewer #1: Yes

Reviewer #2: Yes

5. Review Comments to the Author

Reviewer #1: The study carried out is technically sound with adequate supported research. The argument is well established through literature review and structured methodology. References substantiate the central idea of the paper.

The following are some of the concerns that needs to be reviewed:

In the data provided (uploaded Excel sheet), the classification of word type – Complex/ Non-Complex is confusing. In the study, you have mentioned that;

“The sentences had total 59 words with complex glyphs, such that each sentence had at least four instances of the complex glyphs.”

Please elaborate the premise you used to characterize certain glyphs as complex glyphs.

Additionally, the classification in terms of Word Type requires further justification as in some sentences, 1 or 2 words are classified as ‘Complex’ Word Type while in other sentences, there are NO words that are classified as ‘Complex’ Word Type.

Additional concerns with LINE number:

Line 350-353

“We checked the correlation between the age group of the participants and the RS glyphs written by them using the Pearson's product-moment correlation. The r value was 0.73. The r value between the age group of participants and the TS glyphs written by them was 0.73.”

Please add the interpretation of this particular analysis.

Line 453-460

The reference is on point. So, it can be backed up with further explanation. If possible, please elaborate on how the present study explores the mentioned aspect (Line 460)

Line 131-132 (In-Text citation)

“Studies have indeed shown that the orthographic knowledge for a language form early on in a speaker’s life, and is a function of exposure through education and reading habits (Apel, 2011; Apel et al., 2006; Masterson & Apel, 2010).”

All other references citation in-text has been numbered with the exception of this.

Line 611-612

Lond Rev Educ

The abbreviation of this journal in the journal homepage is mentioned as “The London Review of Education (LRE)”

Journal homepage: https://uclpress.scienceopen.com/collection/2122ff6c-4fcf-4054-a1f0-3be2b1ac2eae#:~:text=The%20London%20Review%20of%20Education,sectors%20and%20phases%20of%20education.

Journal article: https://uclpress.scienceopen.com/hosted-document?doi=10.1080/13603110600574454

Please revise accordingly.

Figure 2

In Figure 2, the schematic representation of the self-paced reading experiment shows the instruction page followed by a sentence in RS. If possible, please add a sentence in TS also.

Reviewer #2: The paper presents a very interesting processing study that has a cultural as well as historical aspect to it. I recommend it for publication. The study looks at whether orthographic knowledge is affected by changes (institutionalised reform) in the standard orthography of a language. Specifically, the paper looks at how the Malayalam script reform in 1971 affected the lexical and sublexical orthographic knowledge in the users of the script. The chronology of the reform allowed the authors to have a pre-reform and a post-reform group. The expectation was that the pre-reform group would take longer to read the reformed script (RS) and the post reform, the traditional script (TS). The participants were also given a production task, specifically a writing task, where they were asked to write down sentences that were dictated to them. I must mention at this point that the script reform was intended as a solution to 'redundant complex grapheme' and an absence of space in the typical manual typewriter of the time. It was predicted that the pre-reform group would choose to write in TS and the post-reform group in RS. Findings showed that both groups were faster in reading RS; and for the writing task, the pre-reform group continued to use TS but the post-reform group wrote in RS.

The results are very interesting, and the analysis provided was enlightening. The reason for both groups being faster was attributed to longer exposure to RS script post 1971. This is taken to imply that lexical orthographic knowledge is flexible depending on exposure. Then the question is, why is this pattern not observed in the writing task? This is where sublexical orthographic knowledge comes into play; the study suggests that sublexical rules ‘seem’ rigid, although their rigidity may be due to a lack of writing practice in the age of digital media. Even if you are a fluent reader of RS, if you learnt TS in school, you might choose TS. This is because writing is also a production task, which may be affected by writing practice. The correlation between group type and script type is taken as further evidence for this type of analysis. An important conclusion that the paper arrives at is the confirmation that orthographic knowledge consists of two independent but related levels: lexical and sublexical orthographic knowledge.

Additional interesting points from the paper:

1. The paper provides a good overview of certain characteristics of Malayalam orthography that can have consequences in terms of language processing, such as complex glyphs. Additionally, it also provides the historical context and contemporary relevance for orthography reforms.

2. The paper also explains how experimental methodologies such as self-paced reading tasks and writing tasks differ; the former is a perception-based task whereas the latter is a production task. The suggestion that a lack of exposure in the form of practice affects production tasks, as they generally incur more processing load is significant.

Points to improve/Suggestions for further research:

In section 6 (lines 414-415), the paper states that the correlation between group type and script type confirms that sublexical knowledge is determined by the script one learns in elementary education. However, does this correlation imply causation or confirmation? This question is significant because the later discussion mentions that sublexical orthographic knowledge could be restructured through continued exposure in the form of writing practice. There seems to be some conflict in clarity there.

In addition to writing being a production skill, it also differs from reading in terms of modality. Writing also has a motor processing component to it in addition to retrieval of orthographic representations. The paper briefly mentions age being a possible factor in writing speed in older participants. Is it possible for this difference in modality to have other consequences when it comes to the choice or preference of script type apart from exposure/writing practice?

It would have been helpful if lexical and sublexical orthographic knowledge were explained using examples/language data.

Similarly, more explanation for certain memory and processing related terms such as retrieval would be helpful for a reader who may be new to the area.

Is it possible that changes made in the script reform were also keeping in view that these revisions would simplify the script in terms of the cognitive load on the user? The paper passingly suggests otherwise. However, do you think it would be right to assume that reforms with such pragmatic intentions would naturally result in a simpler yet more efficient orthographic system?

I would also like to alert the authors of some minor proof-reading errors that need to be corrected.

6. PLOS authors have the option to publish the peer review history of their article (what does this mean?). If published, this will include your full peer review and any attached files.

Reviewer #1: **Yes: **Princy Pappachan

Reviewer #2: **Yes: **Revathi Suresh

---

## [Author Response · Author response to Decision Letter 0]

17 Apr 2023

THE RESPONSE TO REVIEWERS' COMMENTS ARE ATTACHED AS A DOCUMENT 'RESPONSE TO REVIEWERS.DOCX.'

We would like to thank the reviewers for their thought-provoking and helpful remarks. The reviewers have pointed out very valid concerns that needed our attention and we were able to work on those areas. The changes have been made using track change in the manuscript in various places. 

Replies to Reviewer 1 comments:

The following are some of the concerns that needs to be reviewed:

In the data provided (uploaded Excel sheet), the classification of word type – Complex/ Non-Complex is confusing. In the study, you have mentioned that;

“The sentences had total 59 words with complex glyphs, such that each sentence had at least four instances of the complex glyphs.”

Please elaborate the premise you used to characterize certain glyphs as complex glyphs.

Additionally, the classification in terms of Word Type requires further justification as in some sentences, 1 or 2 words are classified as ‘Complex’ Word Type while in other sentences, there are NO words that are classified as ‘Complex’ Word Type. 

ANSWER: Glyphs from TS with conjoined conjuncts and vowel notations attached to the consonants and conjuncts (Fig 1) are classified in the study as complex glyphs. The 50 Malayalam sentences had at least four instances of these complex glyphs out of which words with two or more than two complex glyphs in them were classified as the Complex Word Type. A sentence from the experiment set, thus, would have complex glyphs but not always the Complex Word Type. We have elaborated and rewritten this section in the manuscript (lines 257-269).

Additional concerns with LINE number:

Line 350-353

“We checked the correlation between the age group of the participants and the RS glyphs written by them using the Pearson's product-moment correlation. The r value was 0.73. The r value between the age group of participants and the TS glyphs written by them was 0.73.”

Please add the interpretation of this particular analysis.

ANSWER: There is an inverse correlation between the age group of participants with RS glyphs and a positive correlation with TS glyphs.

We have interpreted the correlation results in the discussion (lines 425-431).

Line 453-460

The reference is on point. So, it can be backed up with further explanation. If possible, please elaborate on how the present study explores the mentioned aspect (Line 460)

ANSWER: We have now elaborated how the study explored Dual Route Theory of Reading aloud (lines 471-479).

Line 131-132 (In-Text citation)

“Studies have indeed shown that the orthographic knowledge for a language form early on in a speaker’s life, and is a function of exposure through education and reading habits (Apel, 2011; Apel et al., 2006; Masterson & Apel, 2010).”

All other references citation in-text has been numbered with the exception of this.

ANSWER: In-text citation has been updated and numbered accordingly (line 133).

Line 611-612

Lond Rev Educ

The abbreviation of this journal in the journal homepage is mentioned as “The London Review of Education (LRE)”

Journal homepage: https://uclpress.scienceopen.com/collection/2122ff6c-4fcf-4054-a1f0-3be2b1ac2eae#:~:text=The%20London%20Review%20of%20Education,sectors%20and%20phases%20of%20education.

Journal article: https://uclpress.scienceopen.com/hosted-document?doi=10.1080/13603110600574454

Please revise accordingly.

ANSWER: Thank you for the feedback. Here the IEEE reference system is automatically formatting the journal article names in an abbreviated form. This necessarily need not be same as the abbreviation of the journal title from their web page. 

Figure 2

In Figure 2, the schematic representation of the self-paced reading experiment shows the instruction page followed by a sentence in RS. If possible, please add a sentence in TS also.

ANSWER: We have added the TS sentence in Figure 2 now (Line 282).

Replies to Reviewer 2 comments:

Points to improve/Suggestions for further research:

In section 6 (lines 414-415), the paper states that the correlation between group type and script type confirms that sublexical knowledge is determined by the script one learns in elementary education. However, does this correlation imply causation or confirmation? This question is significant because the later discussion mentions that sublexical orthographic knowledge could be restructured through continued exposure in the form of writing practice. There seems to be some conflict in clarity there.

ANSWER: It seems that there is a causation between group type and sublexical knowledge, because the age of acquisition of the script seems to govern the script that one uses to write in the production task. However, the two outliers indicate that the script for writing (and hence the sublexical knowledge) is an updatable skill. So, it seems like the relationship that we have discovered is really between sublexical knowledge and exposure/practise of the script. This is further discussed in lines 441-458.

In addition to writing being a production skill, it also differs from reading in terms of modality. Writing also has a motor processing component to it in addition to retrieval of orthographic representations. The paper briefly mentions age being a possible factor in writing speed in older participants. Is it possible for this difference in modality to have other consequences when it comes to the choice or preference of script type apart from exposure/writing practice?

ANSWER: Thank you for the question. It is very relevant to think about the effect of the modality itself when choosing one script over the other. When reading, the script choice is rarely subjective as both the visual and print media overtly choose the script for the viewers/readers. As we see in our results, the reading speed is affected by age. Despite the senile slowdown, the RS was read faster than TS by the elderly population. When it comes to writing, we did not overtly ask the participants to pick any script. Still the elderly picked TS for writing and the young participants chose RS. The speed of writing was not calculated during the task, as we wanted to know which script the participant would select if not explicitly asked to choose between scripts. This is an indication of their sublexical orthographic knowledge (Apel, 2011)

. The effect of modality here could be then the overt and covert choices the participants get to make. The overt choices are mostly a reflection of their lexical orthographic knowledge seen while reading tasks and the covert choices are reflection of their sublexical orthographic knowledge seen in the writing task. Reading, thus, is not a subjective activity in terms of choice of scripts as compared to writing, where the reader consciously or sub consciously gets to choose the script. 

It would have been helpful if lexical and sublexical orthographic knowledge were explained using examples/language data.

ANSWER: We have added examples from language data (lines 126-133). 

Similarly, more explanation for certain memory and processing related terms such as retrieval would be helpful for a reader who may be new to the area.

ANSWER: We added the explanation in the text (lines 181-186).

Is it possible that changes made in the script reform were also keeping in view that these revisions would simplify the script in terms of the cognitive load on the user? The paper passingly suggests otherwise. However, do you think it would be right to assume that reforms with such pragmatic intentions would naturally result in a simpler yet more efficient orthographic system?

ANSWER: The available historic discourse indicates that the government’s decision was made from a policy perspective, specifically to achieve a standard script in Malayalam. We cannot say for certain that this reform has (or has not) resulted in a simpler or efficient orthographic system for the language users, especially the elderly, since it appears that the ease of use is largely practice and exposure-based phenomenon. It must be admitted that the perceptual load for the TS-educated user could decrease due to less phonemic density, but it must be discounted that the cognitive load would for such a user would go up. This is because for a learner who had their elementary education in TS, their schema and sublexical knowledge would be formed by the glyphs and ligatures in TS. An addition of RS glyphs and altered way of writing would increase the cognitive load due to the limited working memory capacity. The newly learnt RS script and the available schema in TS would result in crowding of the cognitive capacity leading to less efficient writing or reading activity by the language user. 

I would also like to alert the authors of some minor proof-reading errors that need to be corrected.

ANSWER: Thank you for pointing this out. We have corrected the proof-reading errors in the manuscript.

---

## [Decision Letter · Decision Letter 1]

2 May 2023

The Effect of script reform on levels of orthographic knowledge: Evidence from alphasyllabary Malayalam scripts

PONE-D-22-29580R1

Dear Dr. Bhargava,

We’re pleased to inform you that your manuscript has been judged scientifically suitable for publication and will be formally accepted for publication once it meets all outstanding technical requirements.

Kind regards,

Iftikhar Ahmed Khan

Academic Editor

PLOS ONE

Additional Editor Comments (optional):

Reviewers' comments:

Reviewer's Responses to Questions

**Comments to the Author**

1. If the authors have adequately addressed your comments raised in a previous round of review and you feel that this manuscript is now acceptable for publication, you may indicate that here to bypass the “Comments to the Author” section, enter your conflict of interest statement in the “Confidential to Editor” section, and submit your "Accept" recommendation.

Reviewer #1: All comments have been addressed

Reviewer #2: All comments have been addressed

2. Is the manuscript technically sound, and do the data support the conclusions?

Reviewer #1: Yes

Reviewer #2: Yes

3. Has the statistical analysis been performed appropriately and rigorously? 

Reviewer #1: Yes

Reviewer #2: Yes

4. Have the authors made all data underlying the findings in their manuscript fully available?

Reviewer #1: Yes

Reviewer #2: Yes

5. Is the manuscript presented in an intelligible fashion and written in standard English?

Reviewer #1: Yes

Reviewer #2: Yes

6. Review Comments to the Author

Reviewer #1: (No Response)

Reviewer #2: (No Response)

7. PLOS authors have the option to publish the peer review history of their article (what does this mean?). If published, this will include your full peer review and any attached files.

Reviewer #1: **Yes: **Princy Pappachan

Reviewer #2: **Yes: **Revathi Suresh

---

## [Editor Report · Acceptance letter]

26 Jul 2023

PONE-D-22-29580R1 

The Effect of script reform on levels of orthographic knowledge: Evidence from alphasyllabary Malayalam scripts 

Dear Dr. Bhargava:

I'm pleased to inform you that your manuscript has been deemed suitable for publication in PLOS ONE. Congratulations! Your manuscript is now with our production department. 

Kind regards, 

on behalf of

Dr. Iftikhar Ahmed Khan 

Academic Editor

PLOS ONE